# Cdc42 interacts with chaperone Ydj1 to enhance its stability and partitioning during asymmetric cell division and aging in yeast

**Pil Jung Kang[1], Hana Mazak[1,2], Sung Sik Lee[3,4], Hay-Oak Park[1]***

1 Department of Molecular Genetics, The Ohio State University, Columbus, Ohio, United States of America, 2 Department of Statistics, The Ohio State University, Columbus, Ohio, United States of America, 3 Scientific Center of Optical and Electron Microscopy, ETH Zurich, Zurich, Switzerland, 4 Department of Biology, Institute of Biochemistry, ETH Zurich, Zurich, Switzerland

* park.294@osu.edu

## Abstract

Cdc42, a small GTPase essential for cell polarity, often becomes hyperactive with age and promotes senescence in yeast and animal cells. Yet, the mechanisms driving its age-related upregulation remain unclear. Here, we show that in budding yeast, Cdc42 accumulates over successive cell divisions and that reducing its levels extends life span. Using microfluidics-assisted live-cell imaging and genetic analysis, we found that Cdc42 is distributed unevenly between mother and daughter cells during division. Daughter cells inherit lower Cdc42 levels, which likely help them remain young. This asymmetric distribution depends on Cdc42's association with and/or release from endomembranes and likely involves Ydj1, a farnesylated Hsp40/DnaJ chaperone anchored to the endoplasmic reticulum. Ydj1 interacts with Cdc42, promoting its stability and proper partitioning during cell division. We propose that ER-bound Ydj1 facilitates the asymmetric distribution of Cdc42, thereby restricting aging to mother cells.

## Introduction

Cdc42 is crucial for establishing cell polarity by regulating cytoskeletal organization and directional growth in organisms ranging from yeast to humans [1,2]. Although it is essential for normal cell function, elevated Cdc42 activity in aged cells—such as budding yeast and hematopoietic stem cells—leads to polarity loss and more symmetric divisions, yet the underlying mechanisms remain poorly understood [3–7]. In budding yeast, polarity is required for asymmetric cell division, which drives mother cell aging while allowing daughter cells to retain their full life span potential. Replicative life span (hereafter, **life span**) is defined as the number of divisions a mother cell undergoes before entering senescence [8]. In multicellular organisms, senescent cells accumulate with age [9], whereas in yeast, senescence is more directly linked

**Data availability statement:** All relevant data are within the paper and its Supporting information files.

**Funding:** This work was supported by the National Institutes of Health/National Institute on Aging (R21-AG060028 to H-OP) and the Ohio State President's Research Excellence Program (to H-OP). The funders had no role in study design, data collection and analysis, decision to publish, or preparation of the manuscript.

**Competing interests:** The authors have declared that no competing interests exist.

**Abbreviations:** ER, endoplasmic reticulum; ERCs, extrachromosomal rDNA circles; GAP, GTPase-activating protein; PM, plasma membrane; ROI, region of interest; SBD I, substrate-binding domain I; VIP, Visible Immunoprecipitation; WT, wild type.

to life span limitation. Aging-associated cellular features—such as increased protein aggregation [10–12], mitochondrial dysfunction [13,14], and loss of rDNA silencing that leads to the formation of extrachromosomal rDNA circles (ERCs) [15,16]—are generally confined to yeast mother cells and are also associated with metazoan aging [17,18]. In yeast, Sir2 silences the rDNA locus and suppresses ERC formation [19], while Fob1 destabilizes rDNA and promotes ERC production [20]. Accordingly, deletion of *SIR2* shortens life span, whereas deletion of *FOB1* extends it relative to wild type (WT) [19,21].

During polarized growth in budding yeast, the "polarisome"—a multi-protein complex crucial for establishing and maintaining cell polarity [22]—facilitates retrograde transport of protein aggregates from buds to mother cells along actin cables [23]. This process also requires Sir2, which promotes proper folding of actin through the chaperonin CCT [23]. Yet, mutations in polarisome components do not completely abolish asymmetric segregation of protein aggregates [24,25], indicating that additional mechanisms contribute to mother-cell-specific aging.

Yeast cells lacking Cdc42-inhibitory polarity cues or Rga1, a Cdc42 GTPase-activating protein (GAP), exhibit shortened lifespans [7,26]. Our previous microfluidic imaging using a Cdc42-GTP biosensor revealed that Cdc42 becomes hyperactivated during aging. Mild overexpression of Cdc42, while not impairing exponential growth, shortens life span and induces characteristics of premature aging [7]. Interestingly, *rga1*Δ cells, which already have elevated Cdc42-GTP levels at young ages, show further increases in Cdc42-GTP as they age. Cells overexpressing *CDC42* ($CDC42_{OV}$) also display a wide range of Cdc42 increases with age [7]. These observations led us to investigate how Cdc42 protein levels are regulated during aging.

Here, we report that Cdc42 accumulates with age and is unevenly segregated between mother and daughter cells during division. This asymmetric distribution, as well as maintenance of Cdc42 levels, depends on its association with Ydj1, a farnesylated Hsp40/DnaJ chaperone. Our findings highlight a critical role for endoplasmic reticulum (ER)-bound Ydj1 in the unequal partitioning of Cdc42 during asymmetric division and likely during aging.

## Results and discussion

### Cdc42 levels increase in aged cells, likely limiting life span

Given that cells lacking the Cdc42 GAP Rga1 show increased Cdc42 activation as they age [7], we examined whether Cdc42 protein levels change across successive cell divisions. We also sought to understand how daughter cells maintain low Cdc42 levels despite strong polarization to the bud tip early in the cell cycle [27]. Using time-lapse imaging of cells expressing Cdc42-mCherry[SW] (a fully functional internal fusion [28]) and Cdc3-GFP (a septin that marks mother and bud compartments), we found that Cdc42-mCherry[SW] fluorescence peaked in small buds early in the cell cycle and then declined as the bud grew, reaching a minimum in large buds or newborn daughter cells (S1 Fig). Based on this observation, we focused our subsequent analyses on cells in late M phase or cytokinesis, when Cdc42 is no longer polarized to the bud and before it localizes to a new bud site.

To compare Cdc42 levels at different ages, we performed microfluidic imaging of WT cells expressing Cdc42-mCherry[SW] (Fig 1A; S1 Movie) and measured mean fluorescence intensity in each cell during its first three divisions ("young mother") and last three divisions before death ("old mother"). These analyses showed that Cdc42 levels increase with cell age (Fig 1B). We next examined how reduced Cdc42 protein levels affect life span using the *cdc42-108(R147A E148A K150A)* allele. This hypomorphic variant produces approximately 50% less Cdc42 protein than the isogenic WT (Fig 1C) but is phenotypically indistinguishable from WT during exponential growth [29]. Lifespan analysis using micro-fluidic imaging showed that *cdc42-108* cells had a median life span about 23.5% longer than WT (Fig 1D). To determine whether this extended life span was due to reduced Cdc42 protein rather than loss of function, we introduced an extra copy of *cdc42-108* into the mutant. This strain, carrying two copies of the *cdc42-108* allele, had a life span similar to WT and shorter than the single-copy *cdc42-108* mutant (Fig 1E). Taken together with the shorter life span of the $CDC42_{OV}$ strain [7], these results suggest that increased Cdc42 levels likely limit life span.

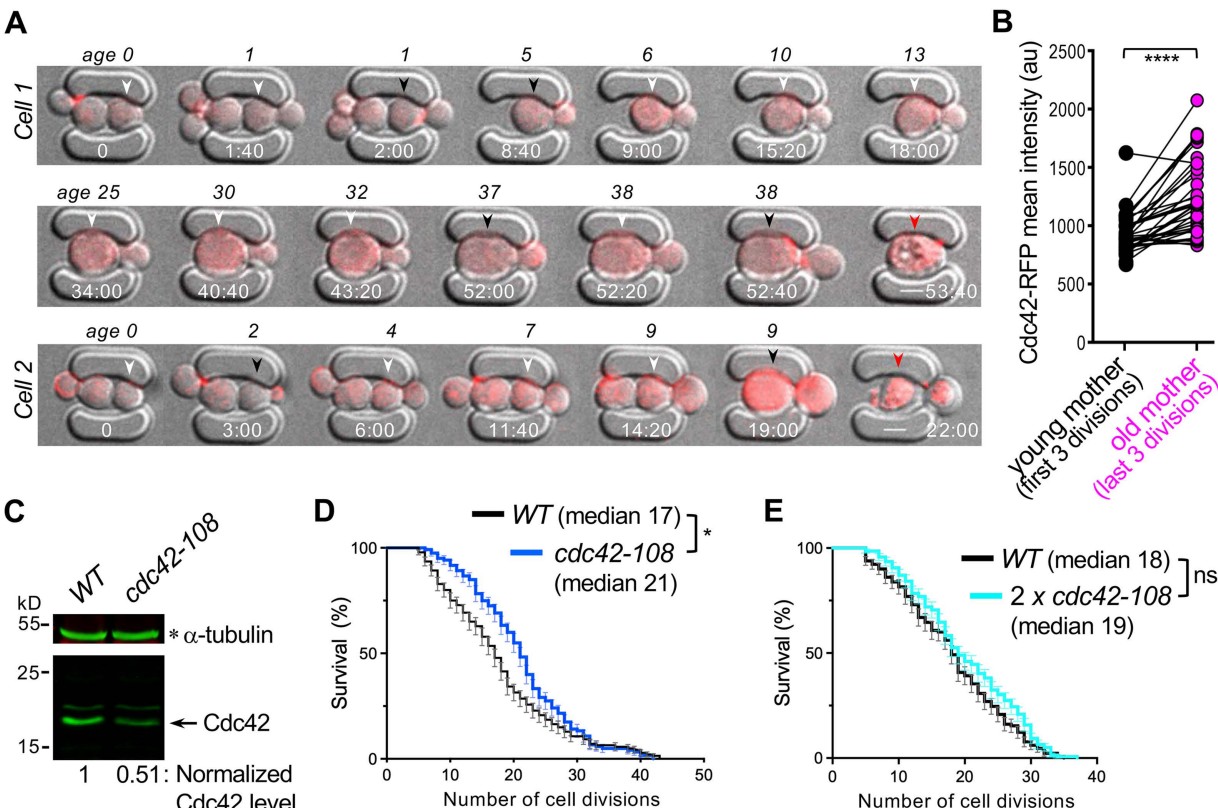

**Fig 1. Cdc42 accumulates in aged cells, and reducing its levels extends life span. A.** Two representative WT cells expressing Cdc42-mCherry[SW] are shown at selected ages and time points (hr: min) from initial loading. White arrowheads indicate cells during or shortly after cytokinesis, when Cdc42 levels are quantified. Black and red arrowheads denote the same cells at other cell cycle stages and cell death, respectively. Scale bar: 3 μm. See S1 Movie. **B.** Mean fluorescence intensity of Cdc42-mCherry[SW] in mother cells at young (first three divisions) vs. old (last three divisions) ages, measured at the large-budded stage (see Fig 1A legend). Each circle represents the average intensity from three divisions of an individual cell. *n* = 37; **** *p* < 0.0001, paired *t*-tests. **C.** Cdc42 proteins in WT and *cdc42-108* cells, grown at 30 °C to mid-log phase, were detected by immunoblotting using a monoclonal anti-Cdc42 antibody, and their levels were normalized to the loading control, α-tubulin. **D.** Percentage of cell survival (mean ± SEM) at each age of WT and *cdc42-108* mutant. Median life span is shown. *n* = 140 (WT) and 120 (*cdc42-108*); *, *p* = 0.0289, Log-Rank test. **E.** Cell survival percentage (mean ± SEM) of WT and a strain with two copies of *cdc42-108* (2 x *cdc42-108*). Median life span is indicated. *n* = 130 (WT) and 139 (*2 x cdc42-108*); ns (not significant), *p* = 0.063, Log-Rank test. The data underlying the graphs can be found in S1 Data.

## Upregulation of Cdc42 likely limits life span independently of rDNA silencing loss

Cdc42 is best known for its role in establishing polarity at the plasma membrane (PM), but it also localizes to intracellular sites at the nuclear envelope and vacuoles [28,30]. Because ERCs associate with the nuclear pore complex and accumulate asymmetrically in mother cells [31,32], we asked whether increased Cdc42 levels might promote aging by interacting with ERCs. Previous studies have shown that distinct terminal bud shapes are linked to specific aging defects: elongated buds correlate with rDNA desilencing and nucleolar destabilization [33,34], whereas round buds are associated with mitochondrial dysfunction [35,36]. Elongated buds appear in ~20% of WT cells near death, but increase to ~60% in *sir2Δ* mutants [37,38], supporting a link between silencing defects and elongated bud morphology.

Using microfluidic imaging of WT cells in two genetic backgrounds, we found that most aging cells had round buds or no buds at terminal stages (Fig 2A). Elongated buds were similarly rare in *CDC42ov* cells, suggesting that elevated Cdc42 does not promote rDNA desilencing. To further test whether Cdc42 upregulation affects rDNA silencing, we analyzed the *cdc42-108* mutation in a *fob1Δ* background, a long-lived strain with reduced ERC formation [21]. Notably, the *cdc42-108 fob1Δ* double mutant exhibited an ~21% increase in median life span relative to *fob1Δ* alone (Fig 2B), indicating that elevated Cdc42 shortens life span via a mechanism independent of rDNA silencing loss.

## Cdc42 is asymmetrically distributed during cell division, likely through its association with, and release from, cellular membranes

We quantified Cdc42-mCherry$^{SW}$ fluorescence in WT mother–daughter pairs across generations. These analyses showed that daughter cells generally inherit lower Cdc42 levels than their mothers, regardless of the mother's age (Fig 3A; see Cell 1, Fig 1A). In rare cases (~19%, $n = 37$), both mother and daughter cells died shortly after division. These short-lived

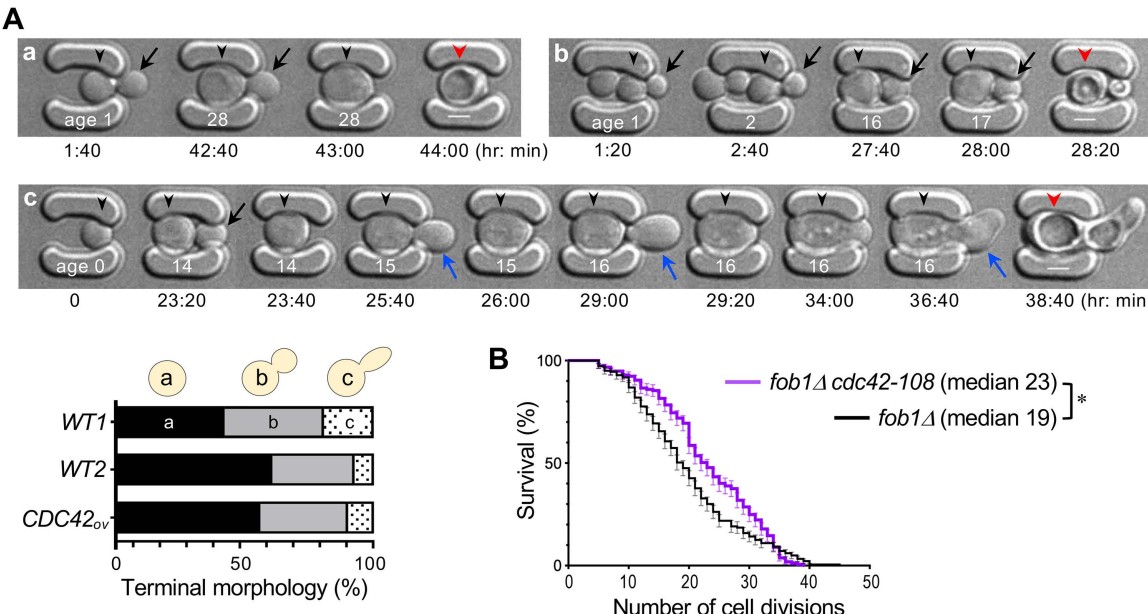

**Fig 2. Cdc42 upregulation limits life span independently of rDNA desilencing. A.** Representative microfluidic images show different cell morphologies during aging: (a) a cell dying in the unbudded state, (b) a cell producing round buds (black arrows) until death, and (c) a cell forming elongated buds (blue arrows). Age and time points are shown in white and black letters, respectively. Scale bar: 3 μm. Quantification of distinct terminal cell shapes is shown below for WT1 (BY4741), and two isogenic strains, WT2 (HPY210) and *CDC42$_{ov}$* (HPY3721) ($n = 105 ~ 110$ per strain). **B.** Cell survival percentage (mean ± SEM) of *fob1Δ* ($n = 183$) and *fob1Δ cdc42-108* mutants ($n = 157$). Median life span is indicated; *, $p = 0.03$, Log-Rank test. The data underlying the graphs can be found in S1 Data.

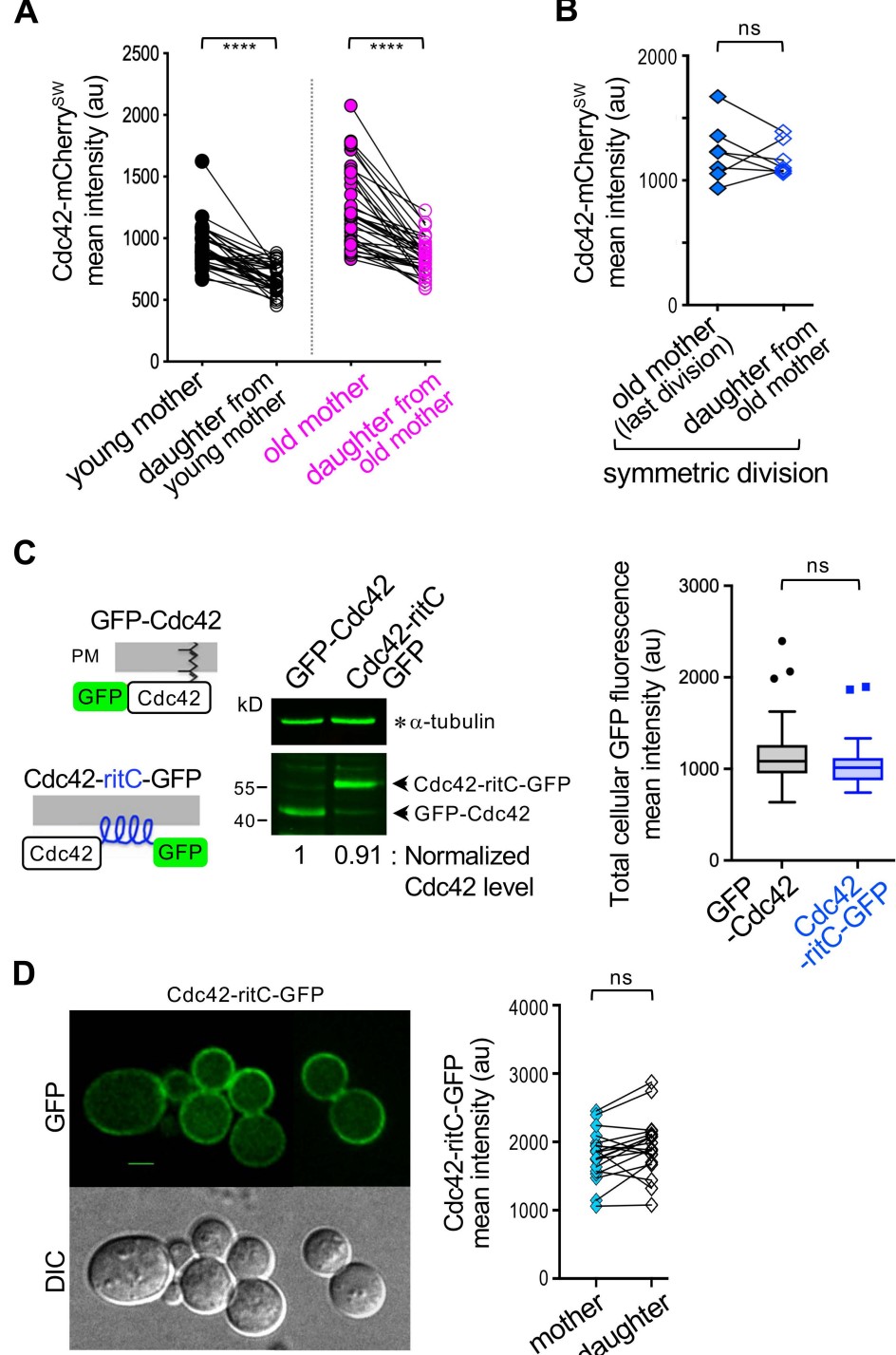

**Fig 3. Asymmetric distribution of Cdc42 during cell divisions. A.** Cdc42-mCherrySW levels in each pair of young and old mothers and their daughters during M-cytokinesis (*n* = 37 pairs, each group). **** *p* < 0.0001 from paired t-tests. Refer to Fig 1B legend. **B.** Cdc42-mCherrySW levels in mother and daughter pairs at the final division preceding death of both cells (see Cell 2 in Fig 1A). *n* = 7 (out of 37 lineages); ns, *p* ≥ 0.05, paired *t* test. **C.** Schematic diagrams of GFP-Cdc42 and Cdc42-ritC-GFP with the Rit amphipathic helix (in blue). Levels of GFP-Cdc42 and Cdc42-ritC-GFP were compared by immunoblotting extracts from cells (expressing each allele as the sole genomic copy) grown at 25 °C to early log phase, using a monoclonal anti-GFP antibody and α-tubulin as a loading control. The mean GFP fluorescence in whole cells (mother and bud combined) of these strains, grown at 25 °C, is

shown in a Tukey plot ($n = 72$ per strain); ns, $p \geq 0.05$, unpaired *t* test. **D.** Localization of Cdc42-ritC-GFP at 25 °C, and quantification in mother and bud compartments during M-cytokinesis. Twenty representative pairs are shown. ns, $p \geq 0.05$, paired *t* test. Scale bar: 3 μm. See S2 Fig and S2 Movie. The data underlying the graphs can be found in S1 Data.

daughter cells inherited high Cdc42 levels comparable to those of their mothers (Fig 3B) and were similar in size to their mothers at division (see Cell 2 at $t = 14{:}20$, Fig 1A). Collectively, these findings suggest that Cdc42 accumulates in mother cells during asymmetric division and that reduced Cdc42 levels in daughters likely help maintain their youthfulness.

To investigate how Cdc42 becomes asymmetrically distributed, we considered two mechanisms: (1) retrograde transport from the bud to the mother cell, as observed for protein aggregate segregation [23]; and (2) age-dependent changes in proteasome activity that influence protein turnover [39]. Retrograde transport occurs during apical growth [40], when Cdc42 is concentrated at the bud tip (see S1 Fig). Although aging may alter proteasome function and Cdc42 turnover, there is little evidence for different turnover rates between young mother cells and their daughters. Neither mechanism is therefore likely to fully explain Cdc42's asymmetric distribution.

We next examined how Cdc42 membrane association affects its distribution, given that Cdc42 localizes to both the PM and endomembranes [27,28,30]. To restrict Cdc42 to the PM, we used the *cdc42-ritC* allele as the sole copy of *CDC42*, in which the C-terminal CaaX motif is replaced with the amphipathic tail of the mammalian Rit GTPase [41]. As previously reported [41], the *cdc42-ritC-GFP* strain grew normally at 24–27 °C but poorly above 30 °C and occasionally formed multiple buds simultaneously (S2 Fig; S2 Movie). Cdc42-ritC-GFP was expressed at levels comparable to those of chromosomally expressed GFP-Cdc42, indicating that protein stability was unaffected (Fig 3C). However, Cdc42-ritC-GFP lost its asymmetric localization, exhibiting similar levels in mother and daughter cells at division (Fig 3D). These observations suggest that Cdc42's association with internal membranes, or its release from the PM, is critical for its asymmetric distribution.

Because *cdc42-ritC-GFP* cells exhibit abnormal morphology and a multi-budding phenotype, we estimated their life span using time-lapse imaging rather than by tracking successive mother cell divisions with microfluidic imaging or micromanipulation. Strikingly, most newborn daughter cells died after only a few divisions at 25 °C (>90%, $n = 100$) (S2 Fig; S2 Movie), supporting the idea that Cdc42 asymmetry is critical for the full life span of daughter cells. However, in addition to defects in endomembrane association, *cdc42-ritC-GFP* cells frequently failed to establish cell polarity after a few divisions, as indicated by the appearance of large, round mother cells that stopped budding (S2 Fig; S2 Movie). This phenotype is likely caused by the slow lateral diffusion of Cdc42-ritC-GFP at the PM or its impaired interaction with GDI (guanine nucleotide dissociation inhibitor) [41,42]. These complex phenotypes make it difficult to directly attribute the mutant's shortened life span to the symmetric distribution of the protein.

### Farnesylated Ydj1 interacts with Cdc42 and supports its asymmetry and stability

Because endomembrane association is likely involved in Cdc42 partitioning, we investigated the role of Ydj1, a farnesylated Hsp40 tethered to the ER [43,44]. Ydj1 is enriched in mother cells via an ER diffusion barrier and retains protein aggregates in aging cells [12]. As *ydj1*Δ mutants grow poorly at 30 °C and are inviable at 37 °C [43,45] (see S3 Fig), we examined Cdc42-mCherry^SW localization in *ydj1*Δ cells at 27 °C, a permissive temperature. In these cells, Cdc42 polarized normally during early cell cycle stages but became symmetrically distributed during division (Fig 4A). We quantified Cdc42-mCherry^SW in mother and bud compartments of large-budded *ydj1*Δ cells (Fig 4B, left) and calculated asymmetry as the log of the mother/bud fluorescence ratio (Fig 4B, right). These analyses indicate that *ydj1*Δ cells are defective in establishing Cdc42 asymmetry during division.

Because farnesylated Ydj1 diffuses more slowly in mother cells than in buds [12], we hypothesized that ER-anchored Ydj1 promotes the asymmetric distribution of Cdc42 during cell division. Consistent with this idea, Cdc42 asymmetry

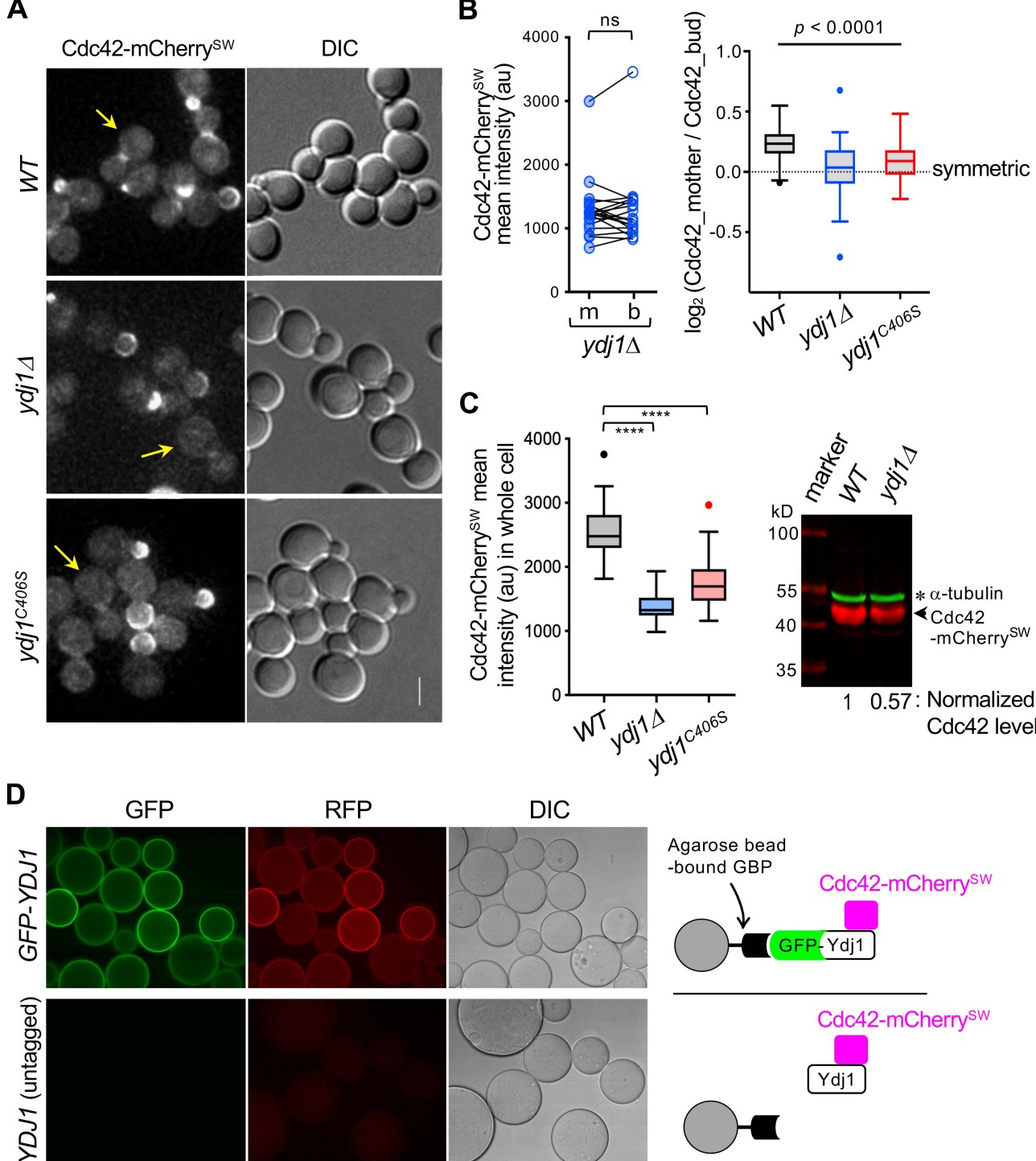

**Fig 4. Farnesylated Ydj1 is required for maintaining Cdc42 levels and its asymmetric distribution.** Localization of Cdc42-mCherry$^{SW}$ in WT and *ydj1* mutants at 27 °C. Arrows mark examples of large-budded cells used for Cdc42 quantification in **B** and **C**. Scale bar: 3 µm. **A.** Cdc42-mCherry$^{SW}$ levels in mother (m) and bud (b) compartments of large-budded *ydj1Δ* cells, grown at 27 °C. (left plot) Mean fluorescence intensities from 19 representative mother-bud pairs are plotted. ns, $p \geq 0.05$, *p*aired *t* test. (right plot) The $\log_2$ mother-to-bud ratio of Cdc42-mCherry$^{SW}$ mean intensity in WT and *ydj1* mutants ($n = 52$ per strain). The dotted line denotes a symmetric distribution of Cdc42 between mother and bud. **** $p < 0.0001$ by one-way ANOVA. See also S3D Fig. **B.** Cdc42 levels in WT and *ydj1* mutants, grown at 27 °C to mid-log *p*hase. Mean fluorescence intensities of Cdc42-mCherry$^{SW}$ in

whole cells (mother and bud combined) are plotted. $n = 57$–60 per strain; **** $p < 0.0001$, unpaired t-tests. Immunoblotting shows Cdc42-mCherry[SW] in each strain, detected using polyclonal anti-RFP antibodies, and α-tubulin, a loading control. See also S3C Fig. **D.** Association of Cdc42-mCherry[SW] with GFP-Ydj1 detected by a visible IP assay (top panel). A control reaction used extracts containing untagged Ydj1 (bottom panel). The data underlying the graphs can be found in S1 Data.

was partially reduced in the farnesylation-defective *ydj1(C406S)* mutant [46], although less severely than in *ydj1Δ* cells (Fig 4B, right). Statistical analysis confirmed a significant association between farnesylated Ydj1 and Cdc42 asymmetry ($p < 0.0001$, one-way ANOVA). In addition, overall Cdc42 protein levels were reduced in *ydj1Δ* and *ydj1(C406S)* mutants compared with WT, as measured by Cdc42-mCherry[SW] fluorescence in whole cells. Consistently, immunoblotting showed an approximately 2-fold reduction in Cdc42 levels in *ydj1Δ* cells (Fig 4C). We next tested whether Cdc42 physically associates with Ydj1 using a visible immunoprecipitation assay [47], which combines immunoprecipitation with fluorescence microscopy. In this assay, lysates from cells expressing GFP-Ydj1 and Cdc42-mCherry[SW] were pulled down with GFP-binding protein-conjugated beads [48]. Fluorescence microscopy showed that both GFP-Ydj1 and Cdc42-mCherry[SW] were retained on the beads, whereas no signal was detected in control samples with untagged Ydj1 (Fig 4D), indicating an in vivo interaction between Cdc42 and Ydj1.

Our study highlights the critical role of Ydj1 in maintaining both Cdc42 protein stability and its asymmetric distribution. Ydj1 cooperates with Hsp70 and Hsp90 chaperones to promote protein folding and stabilization, including protein kinase maturation [49,50]. Previous work showed that the farnesylation-defective *ydj1(C406S)* mutation impairs the stability and maturation of the Hsp90 client Ste11, whereas mutations in substrate-binding domain I (SBD I) do not affect Ste11 accumulation [51]. To test whether this property extends to Cdc42, we examined the *ydj1(L135S)* mutation in the SBD I peptide-binding pocket [52,53]. A low-copy *ydj1(L135S)* plasmid rescued the temperature-sensitive growth defect of *ydj1Δ* mutants, as previously reported [52]. Cdc42 levels in *ydj1Δ* cells carrying either the *L135S* or WT *YDJ1* plasmid were higher than in cells carrying *C406S* or an empty vector (S3C Fig). Quantification of Cdc42 levels in mother and bud compartments separately showed that the *L135S* mutation did not significantly alter Cdc42 asymmetric distribution (WT versus *L135S*, $p > 0.05$), in contrast to the *C406S* mutation (WT versus *C406S*, $p < 0.001$) (S3D Fig). Together with the analysis of *ydj1(C406S)* cells (Fig 4B and 4C), these findings indicate that farnesylated Ydj1 is critical for both Cdc42 stability and its asymmetric distribution, likely via ER tethering.

In contrast to our findings at 27 °C, previous studies reported higher Cdc42 levels in *ydj1Δ* cells than in WT at 30 °C and after shifting to 37 °C [54,55]. The basis for this discrepancy is unclear, but it may reflect temperature-dependent stress responses. At 30 °C or above, *ydj1Δ* mutants exhibit impaired growth and increased expression of Hsp70 and Hsp90 [43,45,50], which could promote Cdc42 stabilization and proper folding at higher temperatures. Thus, the elevated Cdc42 observed in *ydj1Δ* mutants under these conditions may result from indirect compensatory mechanisms rather than the direct consequence of losing Ydj1. Further studies are required to clarify these mechanisms under different conditions.

## Summary and limitations

This study shows that Cdc42 levels increase as yeast cells age and that reducing these levels can extend life span. During cell division, Cdc42 is distributed unevenly between mother and daughter cells. Daughter cells that inherit high Cdc42 levels, comparable to those of their mothers, often die within a few divisions, indicating that proper asymmetric partitioning is crucial for achieving a full replicative life span. However, it remains unclear whether a specific Cdc42 threshold determines proliferative capacity. Mutational and biochemical analyses indicate that Cdc42 asymmetry depends on its association with, and release from, endomembranes, likely mediated through interaction with the farnesylated chaperone Ydj1. Ydj1 is required both to maintain proper Cdc42 levels and to promote its asymmetric distribution. We propose that Ydj1, which diffuses more slowly in mother cells than in buds [12], anchors Cdc42 to the ER in mother cells, leading to its uneven segregation and establishing age-related asymmetry. Many questions remain, including how farnesylated Ydj1

anchors Cdc42 in mother cells and whether Ydj1 cooperates with Hsp90 to promote Cdc42 maturation. This study also did not directly examine how Ydj1 influences this process during aging.

Our data suggest that Cdc42 promotes senescence independently of rDNA silencing loss, but how Cdc42 upregulation drives aging remains unresolved. Cdc42 is not one of the "long-lived asymmetrically retained proteins" typically associated with accumulated cellular damage [56]. Notably, many mother cell-enriched proteins whose depletion extends life span are normally turned over, suggesting that aging or senescence is not solely due to damage accumulation [57,58]. We speculate that repeated asymmetric divisions gradually increase Cdc42 levels in mother cells, ultimately impairing cellular function through aberrant signaling and thereby limiting proliferative capacity. Although this work leaves important questions unanswered, it reveals a chaperone-dependent mechanism for Cdc42 partitioning during asymmetric division that likely contributes to cellular aging and may be conserved in other asymmetrically dividing cells.

## Materials and methods

### Yeast strains, growth conditions, and plasmids

Standard yeast genetics methods, DNA manipulation, and growth conditions were used [59]. Yeast strains were cultured in rich YPD medium (yeast extract, peptone, dextrose) or in the appropriate synthetic medium containing 2% dextrose as a carbon source. All fusion proteins were expressed from their native promoters on the chromosomes, except for GFP-Ydj1, which was expressed from a plasmid for visible immunoprecipitation assays. Unless otherwise indicated, yeast cultures for imaging and protein preparation were grown at 27 °C. The low-copy plasmids pRS315-YDJ1, pRS315-ydj1(C406S), pRS315-ydj1(L135S) (CEN, *LEU2*), and pRS416-GFP-YDJ1 (CEN, *URA3*) were generous gifts from D. M. Cyr (University of North Carolina) and W. Schmidt (University of Georgia) [46,53]. Yeast strains used are listed in S1 Table.

### Microscopy and microfluidic imaging

Cells were grown in an appropriate synthetic medium overnight and then freshly subcultured for 3–4 h in the same medium. Live-cell imaging, including microfluidics-based imaging, was performed using an inverted microscope (Ti-E; Nikon) fitted with a 100×/1.45 NA Plan-Apochromat Lambda oil immersion objective lens, a 60×/1.4 NA Plan Apochromat Lambda oil immersion objective lens, and DIC optics (Nikon), FITC/GFP, and mCherry/Texas Red filters from Chroma Technology, an Andor iXon Ultra 888 electron-multiplying charge-coupled device (EM CCD) (Andor Technology), Sola Light Engine (Lumencor) solid-state illumination, and the software NIS Elements (Nikon).

Static fluorescence images (Figs 3C, 3D, 4A–4C, and S3) were captured using a 100×/1.45 NA objective lens with cells either mounted on a 2% agarose slab or a glass-bottomed dish (MatTek) containing an appropriate synthetic medium, as previously described [42,60]. Due to aberrant cell shapes and a multi-budding phenotype, the life span of *cdc42-ritC-GFP* cells was estimated using long-term time-lapse imaging. Freshly subcultured cells were seeded at a low density in a glass-bottomed dish (MatTek), and images were acquired with a 60×/1.4 NA objective lens using DIC optics every 25 min for approximately 16 h. Representative timepoints are shown in S2 Fig and S2 Movie. The slab or dish was put directly in a stage top chamber (Okolab) set to 27 °C or at room temperature (24–25 °C), as marked in figure legends or figures.

Microfluidics setup and growth conditions are essentially the same as previously described [32], except imaging temperature was maintained at 27 °C. Microfluidic devices were fabricated using polydimethylsiloxane by adopting a design [32] that allows simultaneous imaging of multiple strains through independent passages. Microfluidics-assisted time-lapse imaging was performed using an inverted widefield fluorescence microscope (Ti-E; Nikon) equipped with a 60×/1.4 NA objective lens and DIC optics (see above). In general, bright-field images were recorded every 20 min throughout the entire experiment at multiple XY positions, and fluorescence images were captured for the initial 3–4 h and a few hours after 12, 26, and 50 h, except as indicated. In some cases (Fig 1A), fluorescence images were captured every 20 min with a minimum exposure throughout the entire experiment.

## Image processing and analysis

Images were processed by importing nd2 files using the NIH ImageJ [61] with Bio-Format importer plugin. Mean fluorescence intensities of Cdc42-mCherry$^{SW}$ were quantified by defining a region of interest (ROI) in single-focused z-stack images using the ImageJ oval selection tool only in cells during the late M phase or cytokinesis; i.e., when no polarized Cdc42 signals were detectable at the bud periphery or the incipient bud site (e.g., cells marked with white arrowheads in Fig 1A). Since asymmetric distribution of Cdc42 between mother and bud compartments was established at this cell cycle stage (see S1 Fig), Cdc42 levels at this cell cycle stage were quantified throughout this study. These values were compared between young and old ages (of the same cell lineages) or between a mother and her daughter by averaging the mean intensity of Cdc42-mCherry$^{SW}$ in the first three divisions (young mothers) or last three divisions (old mothers) (Figs 1A, 1B, and 3A). Cdc42 levels were similarly quantified in mother and bud compartments separately in large-budded cells from the exponential growth culture at the indicated temperature (Figs 3D, 4B, and S3). Cdc42 asymmetry was plotted as the $\log_2$ of the Cdc42-mCherry$^{SW}$ fluorescence ratio in each compartment (mother/bud) (Figs 4B and S3). Cdc42-mCherry$^{SW}$ levels in whole cells (mother and bud combined) of WT and mutants were compared by quantifying mean fluorescence intensity using single best-focused z slices, except by defining an ROI covering both a mother and its bud using the ImageJ freehand tool (Figs 4C and S3). Cellular levels of Cdc42-ritC-GFP and GFP-Cdc42 were quantified similarly in whole cells from images of cells in the exponential growth culture at 27 °C (Fig 3C).

To make figures, fluorescence images were deconvolved by the Iterative Constrained Richardson-Lucy algorithm (NIS Elements) and cropped at selected time points, preferentially at cell division or soon after division. The same setting of brightness/contrast was applied to adjust images in multiple panels within the same figure.

## Lifespan estimation

Lifespan was estimated by counting the number of cell divisions observed in microfluidic images starting from initial loading until cell death or senescence. Cell death was identified by abrupt cell shrinkage or lysis seen in DIC images, a sudden and complete loss of fluorescence signal, or the appearance of strong autofluorescence throughout the cell. Cells were considered at senescence or near senescence if their cell cycle length increased sharply (by more than ~6 h) without subsequent division for 8–12 h. Lifespan estimation from microfluidic imaging sometimes underrepresents extremely long-lived cells compared to micromanipulator-based assays, mainly due to rare cell loss or crowding events at later timepoints. Unlike in micromanipulator-based assays, not all initially loaded cells were newborn daughter cells, and thus, the total cell divisions observed were less than the replicative life span. Nonetheless, relative differences in the median number of cell divisions between strains were reproducible, especially since multiple strains were imaged simultaneously using a chamber with multiple flow passages. Cells that died after fewer than five divisions were excluded from life span analysis, as certain daughter cells born from very old mothers did not reach their full life span potential [62].

## TCA protein extraction and immunoblotting

WT and mutant strains were cultured in YPD medium at the temperatures specified in the figure legends and harvested at $OD_{600}$ 0.6–0.8. Whole cell extracts were prepared by precipitation with 10% trichloroacetic acid, as previously described [63]. Protein precipitates were then resuspended in 100 mM Tris (pH 11.0) and 3% SDS, heated at 90 °C for 5 min, and briefly centrifuged to remove cell debris. Proteins were separated by SDS-PAGE on 12.5% polyacrylamide gels, and western blots were visualized using the LI-COR Odyssey system (LI-COR Biosciences, Lincoln, Nebraska) with the following antibodies: mouse monoclonal anti-Cdc42 antibody (clone 28-10) (EMD Millipore, MABN2485), rabbit polyclonal anti-RFP antibodies (Rockland, 600-401-379), mouse monoclonal anti-GFP antibodies (GFP-ID2-s) (DSHB, University of Iowa), and mouse monoclonal anti-alpha tubulin antibody (clone 12G10) (DSHB, University of Iowa). Secondary antibodies used were Alexa Fluor 680 goat anti-rabbit IgG (ThermoFisher Scientific, A32734) or IRDye 800CW goat anti-mouse IgG

(LI-COR Biosciences, 926-32210). Total protein levels in each sample were normalized using Ponceau S-stained membranes. To ensure that α-tubulin, used as a loading control, was present at similar levels relative to total protein in both WT and mutant strains, non-specific cross-reactive bands were also analyzed for comparison.

### Visible immunoprecipitation (VIP) assay

VIP assays were performed using a total of 80 $OD_{600}$ units of HPY4079 (*CDC42-mCherry^{SW} ydj1Δ*) cells transformed with either pRS416-GFP-YDJ1 or pRS315-YDJ1 (control), as previously described [60] with the following modifications. Crude cell lysates were prepared in PBS (PBS, pH 7.4 with 200 mM NaCl, 0.5 mM EDTA) containing a protease inhibitor cocktail (Research Products International) and 0.1 mM PMSF, followed by centrifugation at 10,000*g* for 12 min at 4 °C. Supernatants (S10 fractions) were kept on ice, and pellet fractions (P10) were solubilized using 300 µL of P10 extraction buffer (PBS, pH 7.4, 200 mM NaCl, 0.5 mM EDTA, 2% Triton) by vigorous vortexing and rocking at 4 °C for 1 h. After centrifugation at 16,000*g* for 10 min at 4 °C, the solubilized P10 fractions were collected and mixed with S10 fractions to a final Triton concentration of 0.5%, and then subjected to pull-down assays with GFP-Trap beads (gta-10, Chromotek) at 4 °C. After washing, beads were mounted on an agarose slab and imaged on a Nikon TiE inverted microscope with a 40×/0.75 Plan Fluor objective at room temperature. Static images were captured using DIC, FITC/GFP, or mCherry/Texas Red filters, and single z-slices were selected for figure preparation.

### Statistical analysis

Data analysis was performed using Prism 10 (GraphPad Software). To determine statistical differences between two sets of cell survival data, the Gehan-Breslow-Wilcoxon test and Log-rank (Mantel-Cox) test were used. For comparison of Cdc42 levels between young and old cells in the same lineage or between mothers and their daughters, paired t-tests were performed to determine statistical significance: ns (not significant) for $p \geq 0.05$; *$p < 0.05$, **$p < 0.01$, ***$p < 0.001$, and ****$p < 0.0001$. Statistical significance of the Cdc42-mCherry^{SW} mean intensity differences between mother–bud pairs among WT and mutants was determined by one-way ANOVA analysis. For comparison of Cdc42-mCherry^{SW} in whole cells (mother and bud combined) between WT and mutants, unpaired t-tests were performed. For box graphs, the Tukey method was used to create whiskers (for minimum and maximum data points), a box with a line (for the upper, lower quartiles, and median), and to mark outliers.

### Supporting information

**S1 Fig Quantification of Cdc42-mCherry^{SW} levels in growing buds.** Mean fluorescence intensities of Cdc42-mCherry^{SW} within growing buds are compared. A split septin ring (labeled with Cdc3-GFP) indicates the onset of cytokinesis. Arrows mark the same bud developing and producing a daughter; an arrowhead marks a daughter cell from the previous division. Scale bar: 3 µm. $n = 32$–40 per group; **** $p < 0.0001$, Welch's *t* test. The data underlying the graphs can be found in S1 Data.
(TIF)

**S2 Fig Growth phenotype and time-lapse imaging of *cdc42-ritC* mutant. A.** Growth phenotypes of *cdc42-ritC, cdc42-ritC-GFP,* and isogenic wild-type (WT) strains on YPD plates at 27, 30, and 34 °C. **B.** Time-lapse images of *cdc42-ritC-GFP* cells at 25 °C. Selected time points are shown every 50 min for about 16 hours. Colored asterisks mark daughter cells (d1–d3) originating from the same mother cell (black arrows). Red arrows denote cell death. Note: Daughter cell d2 continued to grow large without typical signs of cell death. See S2 Movie.
(TIF)

**S3 Fig Additional characterization of *ydj1* mutants. A.** Growth phenotypes of *ydj1Δ, ydj1(C406S),* and isogenic WT strains on YPD plates at 27, 30, and 37 °C. **B.** Growth phenotypes of *ydj1Δ* cells carrying either WT YDJ1,

ydj1(C406S), ydj1(L135S), or pRS315 plasmid on SC-Leu plates at 27, 30, and 37 °C. **C.** Mean fluorescence intensities of Cdc42-mCherry[SW] in whole cells (mother and bud combined) of the *ydj1Δ CDC42-mCherry[SW]* strain carrying each plasmid, grown at 27 °C ($n = 78–86$ per strain). **** $p < 0.0001$; *, $p = 0.0333$, by unpaired Welch's $t$ test. **D.** The $\log_2$ mother-to-bud ratio of Cdc42-mCherry[SW] mean intensity (mean ± SEM) in *ydj1Δ* cells carrying each plasmid, grown at 27 °C ($n = 22–40$ per group). $p = 0.0005$ by ordinary one-way ANOVA; ns, $p \geq 0.05$; ** $p < 0.01$; and *** $p < 0.001$ by Welch's $t$ test. The data underlying the graphs can be found in S1 Data.
(TIF)

**S1 Movie. Microfluidic imaging of WT cells expressing Cdc42-mCherry[SW].** Images were captured every 20 min at 27 °C using an inverted widefield microscope (TiE, Nikon) with a 60×/1.4 NA objective lens, and fluorescence images were deconvolved. The movie shows frames for 56 h 20 min of Cell 1, with selected time points shown in Fig 1A. The display rate is 8 frames per sec (fps).
(AVI)

**S2 Movie. Time-lapse imaging of Cdc42-ritC-GFP cells.** Images were captured every 25 min at 25 °C using an inverted widefield microscope (TiE, Nikon) with a 60×/1.4 NA objective lens and DIC optics. The movie shows frames every 50 min for 15 h 50 min. The display rate is 8 frames per sec (fps). An example of cropped images is shown in S2 Fig.
(AVI)

**S1 Table. Yeast strains used in this study.**
(PDF)

**S1 Raw images. Uncropped Western blots for figures.** Full-size, uncropped, and labeled Western blots are shown for the indicated figure panels. The relevant strain genotypes, protein bands, and primary antibodies are labeled. Fig 1C. Cdc42 proteins in wild-type and *cdc42–108* strains. Fig 3C. GFP-Cdc42 and Cdc42-ritC-GFP, expressed chromosomally in respective strains. Fig 4C. Cdc42-mCherry[SW] proteins in wild-type and *ydj1Δ* strains.
(PDF)

**S1 Data. This file contains image quantification data used to generate graphs and statistical analyses shown in the manuscript.** Data for each figure are shown in each sheet.
(XLSX)

## Acknowledgments

We are grateful to M. Peter and the Department of Biology at ETH Zurich for their generous support with microfluidics and for hosting a short-term visit to the Institute of Biochemistry, ETH Zurich. We also thank K. Kozminski, D. Drubin, W. Schmidt, D. G. Cyr, S. Martin, and D. Lew for yeast strains and plasmids.

## Author contributions

**Conceptualization:** Pil Jung Kang, Hay-Oak Park.

**Data curation:** Pil Jung Kang.

**Formal analysis:** Pil Jung Kang, Hana Mazak, Hay-Oak Park.

**Funding acquisition:** Hay-Oak Park.

**Investigation:** Pil Jung Kang, Hana Mazak.

**Methodology:** Pil Jung Kang, Sung Sik Lee.

**Resources:** Pil Jung Kang, Sung Sik Lee.

**Supervision:** Pil Jung Kang, Hay-Oak Park.

**Validation:** Pil Jung Kang, Hay-Oak Park.

**Visualization:** Pil Jung Kang, Hana Mazak, Hay-Oak Park.

**Writing – original draft:** Pil Jung Kang, Hay-Oak Park.

**Writing – review & editing:** Pil Jung Kang, Hana Mazak, Sung Sik Lee, Hay-Oak Park.

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
