## [Editor Report · Decision Letter 0]

8 Jul 2025

Dear Dr Park,

Thank you for submitting your manuscript entitled "Cdc42 Partitioning by Chaperone Ydj1 During Asymmetric Division and Aging in Yeast" for consideration as a Short Report by PLOS Biology.

Your manuscript has now been evaluated by the PLOS Biology editorial staff as well as by an academic editor with relevant expertise and I am writing to let you know that we would like to send your submission out for external peer review.

Once your full submission is complete, your paper will undergo a series of checks in preparation for peer review. After your manuscript has passed the checks it will be sent out for review. To provide the metadata for your submission, please Login to Editorial Manager (https://www.editorialmanager.com/pbiology) within two working days, i.e. by Jul 10 2025 11:59PM.

Kind regards,

Ines

--

Ines Alvarez-Garcia, PhD

Senior Editor

PLOS Biology

---

## [Decision Letter · Decision Letter 1]

10 Sep 2025

Dear Dr Park,

Thank you for your patience while your manuscript entitled "Cdc42 Partitioning by Chaperone Ydj1 During Asymmetric Division and Aging in Yeast" was peer-reviewed at PLOS Biology. Please also accept my sincere apologies for the delay in sending you our decision. The manuscript has now been evaluated by the PLOS Biology editors, an Academic Editor with relevant expertise, and by three independent reviewers.

The reviews are attached below. As you will see, the reviewers find your conclusions interesting, but they also raise several points that would need to be addressed before we can consider the manuscript for publication. Reviewer 1 requests different controls for the Western blots and an additional mutant, along with several minor points. Reviewer 2 notes that the evidence showing how increased Cdc42 levels might contribute to aging needs to be strengthened by considering an alternative hypothesis that the triple mutations in the variant protein disrupts some unknown function that promotes senescence, which could be tested by introducing a second genomic copy of the cdc42-108 allele. This reviewer also thinks that you should clarify whether the Cdc42-ritC derivative has an impact on lifespan. Reviewer 3 finds the observations interesting but still preliminary and mentions that the different cells used in the study should be characterized further to confirm the findings.

In light of the reviews, we would like to invite you to revise the work to thoroughly address the reviewers' reports. Given the extent of revision needed, we cannot make a decision about publication until we have seen the revised manuscript and your response to the reviewers' comments. Your revised manuscript is likely to be sent for further evaluation by all or a subset of the reviewers.

**IMPORTANT - SUBMITTING YOUR REVISION**

3. Resubmission Checklist

a) *PLOS Data Policy*

b) *Published Peer Review*

Sincerely,

Ines

--

Ines Alvarez-Garcia, PhD

Senior Editor

PLOS Biology

Reviewers' comments

Rev. 1:

The manuscript "Cdc42 Partitioning by Chaperone Ydj1 During Asymmetric Division and Aging in Yeast" by Kang et al. shows that Cdc42 partitions asymmetrically during growth and its association with the chaperone Ydj1 is important for this asymmetry that is has a role in cellular aging. The authors show that Cdc42 levels increase during aging, which is likely to be independent of rDNA silencing. Furthermore, they go on to show that this asymmetric partitioning of Cdc42 is not observed when this GTPase is constitutively targeted to the plasma membrane and that such asymmetric partitioning requires the farnesylated chaperone Ydj1. This brief report is clearly written, the data are convincing and the results are likely to be of interest to the broad audience of Plos Biology. There are some minor issues and details, which should be addressed to improve clarity and accessibility:

Minor issues:

1) Some of the graph axes titles are unclear, including Fig. 1 'young and old ages' and Fig. 3A-C, in which it is not clear what the difference is between mother/daughter and young or old mother/daughter.

2) In the western blots, it is not clear that tubulin (which also may change with cell age) is the best control for normalizing protein levels. If other controls are not feasible this point should be mentioned.

3) From Fig. 3 it is concluded that Cdc42 endomembrane assocation is critical, but another interpretation would be that association/dissociation from the membrane is what is critical and this should be mentioned.

4) This is likely to be a semantic point; does lower levels of Cdc42 in daughters 'help them rejuvinate' (in abstract) or rather and more likely keep them young/slow down aging.

5) Is the effect of the ydj1 mutant (Fig. 4) due to a lack of an 'anchor' in the mother cell or an overall reduction of Cdc42 levels. The latter possibility seems to be a simpler explanation. This could be addressed by a repressible mutant or degron mutant, but I appreciate this is beyond the scope of this short report and hence should be mentioned.

6) It is unclear if cdc42-ritC is the sole copy of cdc42, if this is the case, does this strain have an aging defect, as would be expected?

7) In Fig. 4B, it seems that a ratio of mother/bud signals would be more appropriate than the bud value substracted from the mother signal.

Rev. 2:

This work reports that replicative lifespan in budding yeast depends on the asymmetric partitioning of Cdc42, a key regulator of cell polarity. Earlier findings suggested a connection between Cdc42 hyperactivation and premature aging in yeast. Here, the authors explore this connection using time-lapse fluorescence microscopy in microfluidic chambers, which allowed them to visualize the localization and accumultation of Cdc42 during numerous cell divisions of aging cells. They find that, although Cdc42 is enriched at the growing tips of daughter cells (bud) early in the division process, prior to cytokinesis Cdc42 becomes enriched in mother cells, and hence daughter cells inherit lower levels. Old mother cells accumulate increasing levels of Cdc42, suggesting that high levels may contribute to senescence and low levels in new daughter cells is rejuvenating. In support of this hypothesis, mutations in Cdc42 that reduce its levels confer increased lifespan. Further results show that the mother-bud asymmetry of Cdc42 depends on the ER-tethered chaperone protein, Ydj1, and that Cdc42 physically interacts with Ydj1 in cell extracts. Collectively, the findings reveal a surprising partitioning phenomenon for Cdc42 along with a potential underlying mechanism and physiological impact on yeast lifespan.

Overall, this manuscript provides thought-provoking findings and valuable new insights for the field. It fulfills the criteria for Short Reports format papers, in that it provides a concise presentation of novel and intriguing findings from a limited set of experiments. The experiments incisively discriminate among possible models, the data and analysis are high quality, and the interpretations are logical. I don't have any serious concerns or reservations about the work. As discussed in item #1 below, one minor weak point is the degree to which the level of Cdc42 (or the failure to partition it into the mother cell) can be regarded as a causal factor in aging. This is not a fatal flaw, but rather an area for potential improvement that might strengthen the overall conclusions.

Specific points:

1. To address how increased Cdc42 levels might contribute to aging, the authors use a mutant allele, cdc42-108, to reduce Cdc42 levels by roughly a factor of 2. The results clearly show that this allele increases lifespan (Figs 1D, 2B), which is consistent with the hypothesis that elevated Cdc42 levels play a causal role in aging. This is a reasonable interpretation. However, an alternative hypothesis is that the triple mutations in this variant protein (R147A E148A K150A) disrupt some unknown function that promotes senescence. A possible way to distinguish between these alternatives would be to introduce a second genomic copy of the cdc42-108 allele. If the 1x mutant allele conferred a longer lifespan than the 2x mutant allele (which might show a similar lifespan as the WT allele), it would more convincingly argue that the lifespan difference is dictated by different levels rather than by different functionality. Such an approach could strengthen the causality argument, but it is not essential.

2. The authors did not report whether the Cdc42-ritC derivative, which disrupts the asymmetry in mother-bud distribution, had an impact on lifespan. This could potentially address whether the normal asymmetric partitioning has a causal effect on aging. Conceivably any interpretaion could be confounded if the ritC fusion substantially alters total Cdc42 protein levels. It was not clear if the results address this, as the fluorescent tag is different for Cdc42-ritC (GFP) than for other experiments (mCherry). Some direct commentary on this issue seems worthwhile.

3. Page 5: Regarding Fig 4C, the authors state that "Cdc42-mCherrySW fluorescence was diminished in both mother and bud compartments of ydj1 mutant cells". The data don't seem to support this statement. Namely, the statement seems to imply that the fluorescence was measured separately in both mother and bud, and that the levels were reduced in each compartment. But Fig 4C does not analyze mother and bud separately, and instead measures them combined together (from the legend: "...intensities in the whole cells [including both mother and bud compartments]..."). I suspect the confusion is unintentional, and perhaps it could be fixed simply by rewording the main text to say "...the total cellular Cdc42-mCherrySW fluorescence (mother and bud combined) was diminished in ydj1 mutant cells".

Rev. 3:

In this report the authors show that Cdc42 asymmetrically distributed in aging cells and enhanced Cdc42 levels is associated with replicative aging. This is an important observation since enhanced Cdc42 levels has been observed in different aged cells types. Increased Cdc42 levels have been shown to shorten life span right from yeast to animals. However, it is not clear how this is mediated at the molecular level. The authors show that the asymmetric distribution is dependent on farnesylated Ydj1, an Hsp40/DnaJ chaperone tethered to the endoplasmic reticulum. While this is an interesting observation, it is too preliminary in its current form. Additional characterization of this observation will make these findings more convincing. The major issues are mentioned below.

Major points

1. The authors show that Cdc42-RitC fails to asymmetrically distribute between the two daughter cells. If asymmetric Cdc42 distribution continues to aging, these cells should not age as rapidly. How do the Cdc42-RitC cells age? Given that Cdc42-RitC is remains in the plasma membrane, if the cells age slowly then it will indicate that the enhanced endomembrane Cdc42 is needed for aging. If they age the same as control cells then only the overall Cdc42 levels matter and not its location.

2. How do ydj1� cells age? Do they show a phenotype similar to increased life span in fob1� mutants. Would a fob1�ydj1� show an enhanced life span while the ydj1�cdc42-108 double mutant would be epistatic? If ydj1� cells age slowly, can this be rescued with Cdc42 overexpression?

3. The enhanced Cdc42 in aged cells could be due to a stress response similar to that observed in cells at high temperature. While the authors present data that ydj1� mutants show decreased levels of Cdc42 at low temperatures, that fact that these mutants show enhanced CDc42 levels at high temperature is confusing and needs further investigations. Is it possible to make a ydj1�hsp90 double mutant? Would this cell show diminished level of Cdc42 even at high temperatures.

Minor points

1. The authors state that the fob1� mutant is long lived. However, the survival curve of fob1� cells in fig2B show shorter live span that the WT cells in fig 1D.

---

## [Decision Letter · Decision Letter 2]

19 Jan 2026

Dear Dr Park,

Thank you for your patience while we considered your revised manuscript entitled "Cdc42 Partitioning by Chaperone Ydj1 During Asymmetric Division and Aging in Yeast" for publication as a Short Report at PLOS Biology. This revised version of your manuscript has been evaluated by the PLOS Biology editors, the Academic Editor and the three original reviewers.

Based on the reviews (attached below), we are likely to accept this manuscript for publication, provided you satisfactorily address the remaining points raised by Reviewer 3. Please also make sure to address the data and other policy-related requests stated below my signature.

In addition, we would like you to consider a suggestion to improve the title:

“Cdc42 interacts with chaperone Ydj1 to enhance its stability and partitioning during asymmetric cell division and aging in yeast”

We expect to receive your revised manuscript within two weeks.

*Published Peer Review History*

*Press*

Sincerely,

Ines

--

Ines Alvarez-Garcia, PhD

Senior Editor

PLOS Biology

DATA POLICY:

Many thanks for providing the data underlying the graphs shownin the figures. Please also ensure that figure legends in your manuscript include information on where the underlying data can be found. For example, you can add at the end of the corresponding figure legends the following sentence: "The data underlying the graphs can be found in S1 Data."

CODE POLICY

Per journal policy, if you have generated any custom code during the course of this investigation, please make it available without restrictions. Please ensure that the code is sufficiently well documented and reusable, and that your Data Statement in the Editorial Manager submission system accurately describes where your code can be found. More information on our Code Policy, what and how to share can be found here: https://journals.plos.org/plosbiology/s/code-availability

Reviewers' comments

Rev. 1:

The authors have addressed all the issues that I have raised as well as addressing comments of the two other reviewers.

Rev. 2:

The authors have addressed all of my prior comments in a fully satisfactory way. In particular, the comparison of 2x vs 1x copies of the cdc42-108 allele provides valuable additional support for the model that Cdc42 protein levels have a causal effect on aging. I also appreciated the efforts to investigate the lifespan effects of the cdc42-ritC variant, as well as the explanations for the complexities posed by the morphological defects in this strain. In addition, I reviewed the responses to comments from the other referees, and I found them to be substantial and thorough. Overall, I am completely satisfied by the revisions and I enthusiastically recommend publication.

Rev. 3:

For a brief report I do not expect mechanistic insights and I recognize the experimental limitations as described here. However, even in a brief report, the claims need to be properly validated. The authors claim in a subtitle that "Cdc42 is asymmetrically distributed during cell division through its association with endomembranes". However, this statement has not been tested. What the authors do show is that stably restricting Cdc42 to the plasma membrane (Cdc42-RITC) leads to symmetric Cdc42 distribution between mother and daughter buds. It is not shown whether this symmetry is due to the lack of Cdc42 dynamics at the plasma membrane, the lack of exchange with the GDIs or the lack of localization to the endomembrane. Moreover, that fact that Cdc42 levels drop in ydj1 mutants indicate that asymmetry could be either due to Ydj1 mediated Cdc42 stability or due to Ydj1 anchoring Cdc42 to the ER. Thus the claim that endomembrane associated Cdc42 leads to asymmetry is not substantiated. The authors show properly address these limitations. With this change this manuscript will be suitable for publication.

---

## [Editor Report · Decision Letter 3]

27 Jan 2026

Dear Dr Park,

Thank you for the submission of your revised Short Report entitled "Cdc42 interacts with chaperone Ydj1 to enhance its stability and partitioning during asymmetric cell division and aging in yeast" for publication in PLOS Biology. On behalf of my colleagues and the Academic Editor, Sophie Martin, I am delighted to let you know that we can in principle accept your manuscript for publication, provided you address any remaining formatting and reporting issues. These will be detailed in an email you should receive within 2-3 business days from our colleagues in the journal operations team; no action is required from you until then. Please note that we will not be able to formally accept your manuscript and schedule it for publication until you have completed any requested changes.

PRESS

Sincerely,

Ines

--

Ines Alvarez-Garcia, PhD

Senior Editor

PLOS Biology
